# Structural basis of spike RBM-specific human antibodies counteracting broad SARS-CoV-2 variants

Kiyomi Shitaoka [1,8], Akifumi Higashiura[2,8], Yohei Kawano[1,8], Akima Yamamoto[2], Yoko Mizoguchi[3], Takao Hashiguchi [4], Norihisa Nishimichi[1,5], Shiyu Huang[1], Ayano Ito [1], Shun Ohki[1], Miyuki Kanda[6], Tomohiro Taniguchi[7], Rin Yoshizato[1], Hitoshi Azuma[1], Yasuo Kitajima[1], Yasuyuki Yokosaki[1,5], Satoshi Okada [3], Takemasa Sakaguchi [2] & Tomoharu Yasuda [1✉]

The decrease of antibody efficacy to mutated SARS-CoV-2 spike RBD explains the breakthrough infections and reinfections by Omicron variants. Here, we analyzed broadly neutralizing antibodies isolated from long-term hospitalized convalescent patients of early SARS-CoV-2 strains. One of the antibodies named NCV2SG48 is highly potent to broad SARS-CoV-2 variants including Omicron BA.1, BA.2, and BA.4/5. To reveal the mode of action, we determined the sequence and crystal structure of the Fab fragment of NCV2SG48 in a complex with spike RBD from the original, Delta, and Omicron BA.1. NCV2SG48 is from a minor $V_H$ but the multiple somatic hypermutations contribute to a markedly extended binding interface and hydrogen bonds to interact with conserved residues at the core receptor-binding motif of RBD, which efficiently neutralizes a broad spectrum of variants. Thus, eliciting the RBD-specific B cells to the longitudinal germinal center reaction confers potent immunity to broad SARS-CoV-2 variants emerging one after another.

[1] Department of Immunology, Graduate School of Biomedical and Health Sciences, Hiroshima University, Hiroshima, Japan. [2] Department of Virology, Graduate School of Biomedical and Health Sciences, Hiroshima University, Hiroshima, Japan. [3] Department of Pediatrics, Graduate School of Biomedical and Health Sciences, Hiroshima University, Hiroshima, Japan. [4] Laboratory of Medical Virology, Institute for Frontier Life and Medical Sciences, Kyoto University, Kyoto, Japan. [5] Integrin-Matrix Biomedical Science, Translational Research Center, Hiroshima University, Hiroshima, Japan. [6] Collaborative laboratory of Liquid Biopsy, Graduate School of Biomedical and Health Sciences, Hiroshima University, Hiroshima, Japan. [7] Division of General Internal Medicine and Infectious Diseases, Hiroshima Prefectural Hospital, Hiroshima, Japan. [8]These authors contributed equally: Kiyomi Shitaoka, Akifumi Higashiura, Yohei Kawano. ✉email: yasudat@hiroshima-u.ac.jp

The antigenic drift of the SARS-CoV-2 RNA virus causes immune evasion through the accumulated mutations in the spike (S) protein, especially in the receptor binding domain (RBD), which reduces the effectiveness of antibodies elicited by vaccination or viral infection[1,2]. The emergence of the SARS-CoV-2 Beta (B.1.351) and Delta (B.1.617) variants raised concern that the progress of antigenic drift actually enhanced transmissibility, severity, and mortality of the disease[3]. In November 2021, the Omicron (B.1.1.529) variant was detected and rapidly spread worldwide with derivative lineages[4,5]. A striking feature of Omicron is a large number of mutations in the S protein which causes a substantial threat to the efficacy of the current COVID-19 vaccine and antibody therapies[6]. The Omicron variant BA.1 has as many as 34 mutations in the S protein compared to the original SARS-CoV-2 strain, Wuhan-Hu-1. Fifteen of them are accumulated in the receptor-binding domain (RBD) which is a primary target of neutralizing antibodies produced after infection or vaccination, including nine mutations located in the receptor-binding motif (RBM), an RBD subdomain that interacts directly with the host receptor ACE2[7]. While recent studies indicated a reduced sensitivity of Omicron variants to developed therapeutic monoclonal antibodies (mAbs) and COVID-19 convalescent sera[7–10], little is known about the properties of antibodies that can neutralize broadly diversified SARS-CoV-2 variants and how such antibodies can be generated and maintained by way of immunization or infection. Furthermore, it is unclear how fast the host immune system can generate broadly neutralizing antibodies (bnAbs) from the primary immunization and to what extent variants can be neutralized by those bnAbs.

A variable region of the antibody consisting of a pair of immunoglobulin heavy chain (HC) and light chain (LC) includes framework regions (FWs) and complementarity determining regions (CDRs) in which somatic hypermutations (SHMs) are introduced in different frequencies[11]. The CDRs contain the loop which creates a specific interface with an antigen accounting for the specificity of the antibody[12,13]. SHMs are preferentially found in the CDRs which generally increases the affinity against antigen[14]. Germinal centers (GCs) arising in lymphoid follicles after the infection or immunization are composed of B cells undergoing rapid clonal expansion and selection thereby contributing to antibody diversification and affinity maturation over weeks to months[15–18]. Indeed, Muecksch et al. showed that SHMs acquired in the months after SARS-CoV-2 infection endow some antibodies specific for RBD of S protein with greater neutralization potency and breadth[19]. Those GC B cells will further differentiate into either memory B cells or plasma cells that contribute to immunological memory maintained over a long period and participate in recall responses to antigens. Because SHMs introduced in GC reaction play a critical role in altering binding properties of antibodies according to viral antigenic drift[14], the investigation of antibodies produced in individuals who acquired universal antibodies to antigenic drift will help to understand a human immune system to fight against viral mutations emerging one after another, which closely relate with the development of ideal vaccines and mAbs for therapeutic use.

To investigate the mechanism of action and structural basis of bnAbs from COVID-19 convalescent individuals infected with early SARS-CoV-2 strain (B.1.1, 2020/4) and experienced a long-term GC reaction, here we identified and characterized bnAbs obtained from those donors predetermined neutralizing activity against broad variants including Omicron.

## Results

### Identification of broadly neutralizing antibodies against SARS-CoV-2 variants.
To determine the donors who acquired broadly neutralizing antibodies, we tested sera from convalescent

18 subjects sampled at 8–55 days post-diagnosis of early D614G strain (B.1.1) infected for the first time without vaccination (Supplementary Table 1). Wuhan-Hu-1 S-trimer-specific IgM, IgG, and IgA levels were determined in the different periods of hospitalization, 8–10, 11–15, and 17–55 days from PCR diagnosis with uninfected healthy donor controls. S-trimer-specific antibody level was the highest in IgG isotype from convalescent people after 17–55 days of hospitalization (Supplementary Fig. 1a). Neutralization assay to authentic D614G strain of SARS-CoV-2 (B.1.1) indicated that sera collected over 17 days from diagnosis contain highest neutralization activity (Supplementary Fig. 1b). We observed that oxygenation treated severe patients rather than mild or moderate convalescent donors acquire the highest neutralization activity, in contrast, no clear correlation between donor age and neutralizing activity (Supplementary Fig. 1c, d). Thus, to gain higher neutralizing activity, a time period over 17 days was required for convalescent COVID-19 patients.

The first Omicron variant BA.1 has 15 mutations in the RBD domain including 9 mutations in the RBM, an RBD subdomain that interacts directly with the host receptor ACE2 (Supplementary Fig. 2a). We subsequently examined the neutralization activity against pseudovirus expressing S protein from Wuhan-Hu-1, Delta (B.1.617.2), or Omicron (B.1.1.529.1/BA.1) on the viral membrane. As previously reported[7–10], we confirmed a mild reduction in neutralizing potency against Delta but a substantial reduction against Omicron in all convalescent sera (Fig. 1a). While neutralizing activity elicited by infection of early D614G strain (B.1.1) largely fails to inhibit Omicron, we noticed that some convalescent individuals exhibit the neutralizing activity even though at the lower level. We found two donors (NCV1 and NCV2) showing relatively higher neutralization activity to Omicron (Fig. 1a). The clinical characteristics of convalescent subjects including these two patients are summarized in Supplementary Table 1. Both patients are highly aged (NCV1, 93-year-old; NCV2, 87-year-old) and hospitalized for 55 days because of severe illness. Peripheral blood mononuclear cells from donors hospitalized for more than 17 days (NCV1, 2, 4, 7, and 8) were served for single-cell sorting and processed to produce mAbs (Supplementary Fig. 2b). The S-trimer binding CD19+IgD− isotype switched live memory B cells were single-cell sorted and we obtained full-length paired IgH and IgL chains with the original IgG isotype and its sequence. We recovered 52 sequences from NCV1 and NCV2 together with 50 sequences from control samples (NCV4, 7, and 8) to produce monoclonal IgG antibodies binding S-trimer. Among obtained 102 human IgG sequences, IGHV3-30, IGKV1-5, or IGKV3-20 occupied more than 10% of the whole sequences (Supplementary Fig. 2c). These three are previously reported as the predominant V genes used in SARS-CoV-2 S-specific antibodies[20], indicating that approach allows us to obtain common rearrangements found in potent neutralizing SARS-CoV-2 mAbs from humans. IGHV3-30 and IGKV1-5 were also highly representative in NCV1 and NCV2 (Supplementary Data 3). Of the 102 IgG mAbs sorted by the S-trimer, IgG1, IgG2, and IgG3 subclasses were 86%, 6%, and 8%, respectively, while the kappa and lambda chains are comparably used (Fig. 1b). Thus, IgG1 has dominantly generated against S protein as previously reported[21]. Notably, 77% of obtained mAbs acquired SHMs with variable mutation numbers (Fig. 1c). The amino acid replacement profile showed accumulated mutations in CDR1 and CDR2 more than FR regions in both VH and VL, indicating that most S-trimer-specific antibodies were an outcome of affinity-based selection in the GC (Fig. 1d). To find neutralizing mAbs effective to SARS-CoV-2, we carried out antibody neutralization of the authentic SARS-CoV-2 using VeroE6 cells expressing the transmembrane serine protease TMPRSS2. As a result, we identified four mAbs,

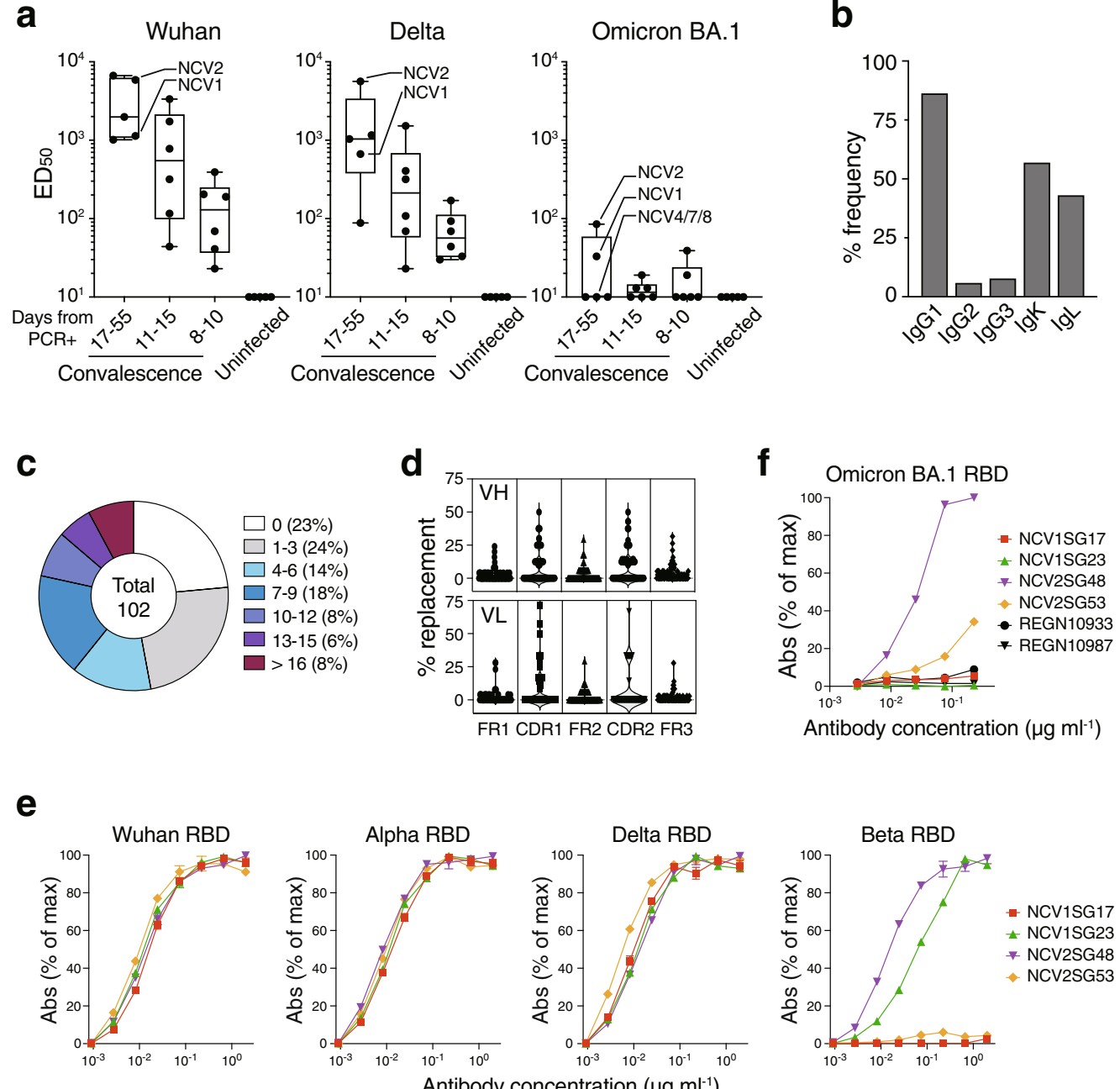

**Fig. 1 Isolation of broadly reacting mAbs from convalescence patients of early SARS-CoV-2 strain. a** Serum neutralizing antibody titers (ED$_{50}$) against SARS-CoV-2 pseudovirus of Wuhan, Delta, or Omicron BA.1. Serum from early COVID-19 convalescent individuals infected first time with B.1.1 (D614G) strain without vaccination or uninfected/non-vaccinated healthy donors were evaluated. COVID-19 convalescent individuals (n = 17) are subgrouped by hospitalization period from PCR results. Subjects used for mAb production (NCV1, 2, 4, 7, 8) were indicated. Each box indicates the median and 25–75 percentile with min to max whiskers. **b**–**d** Sequence of generated 102 mAbs from NCV1, 2, 4, 7, and 8 are analyzed. **b** Frequency of IgG subclasses and light chain isotypes. **c** Distribution of VH and VL amino acid mutation numbers. **d** Percentages of amino acid replacement in FR and CDR region of VH and VL. FR, framework region; CDR, complementarity-determining region. **e**, **f** Binding properties of indicated mAbs against RBDs of SARS-CoV-2 Wuhan-Hu-1 or indicated variants. Data are mean ± SD of technical triplicates. Abs, absorbance.

NCV1SG17, NCV1SG23, NCV2SG48, and NCV2SG53, all of which neutralized D614G authentic SARS-CoV-2 at lower than 1 μg/ml of IC$_{50}$ value (Supplementary Fig. 2d). These four mAbs also neutralized Delta variant, notably NCV2SG48 and NCV2SG53 neutralized Delta virus at lower than 1 μg/ml indicating these mAbs are resistant to antigenic drift of the virus.

**Binding and neutralizing properties of identified mAbs against SARS-CoV-2 variants**. To evaluate whether identified neutralizing

mAbs are effective to Omicron and other variants, we first tested mAbs for binding to various RBD proteins because neutralizing antibodies are known to bind RBD to block interaction with ACE2 or viral entry to target cells. As expected, all four mAbs bound specifically with high affinity to monomeric Wuhan-Hu-1 RBD, and that binding affinity (K$_D$) was 2.5 to 6.1 nM (Supplementary Fig. 3a). Next, we evaluated the binding ability of four mAbs, NCV1SG17, NCV1SG23, NCV2SG48, and NCV2SG53, with mutated RBDs. All four mAbs bound to Alpha and Delta at a similar level with Wuhan

(Fig. 1e). Amino acid substitutions of RBD in SARS-CoV-2 variants used in this study are summarized in Supplementary Fig. 3b. We also included variants acquired point mutation only at K417, L452, or E484, which are known to decrease sensitivity to antibody neutralization[22,23]. RBD binding of all four mAbs was not affected by those mutations (Supplementary Fig. 3c). Of note, both NCV1SG23 and NCV2SG48 mAbs showed binding to all variants including, Alpha, Beta, Kappa, Delta, Delta plus, Lambda, and Omicron BA.1, except for NCV1SG23 which lost binding to Omicron BA.1 (Fig. 1f and Supplementary Fig. 3c). As previously reported[7–9], REGN10933 (Casirivimab) and REGN10987 (Imdevimab) mAbs lost binding to Omicron BA.1 (Fig. 1f).

To confirm the neutralizing ability of NCV1SG17, NCV1SG23, NCV2SG48, and NCV2SG53 against SARS-CoV-2 variants, we carried out a pseudovirus neutralization assay. As expected from the binding results, all four mAbs showed neutralization activity against Wuhan-Hu-1, D614G, Alpha, Delta, Delta plus, and Kappa pseudoviruses (Fig. 2a). While NCV1SG17 and NCV2SG53 did not show neutralization of Beta and Omicron variants, those mAbs neutralized most of the other variants. NCV1SG23 neutralized more broad variants except for BA.1 and BA.2.12.1. Notably, NCV2SG48 neutralized all SARS-CoV-2 variants including BA.1, BA.2, BA.2.12.1, and BA.4/5 (Fig. 2a, b). Neutralization of NCV2SG48 against authentic Omicron BA.1 virus was also confirmed (Supplementary Fig. 2d). Although NCV2SG48 showed a 13-fold decrease in neutralization activity against BA.1 compared to parental D614G (Fig. 2b), 798 ng/ml of $IC_{50}$ value to BA.1 is equivalent level with reported neutralizing antibody Sotrovimab derived from S309 mAb isolated from SARS-CoV infected individual, which inhibits BA.1 by 260-917 ng/ml of $IC_{50}$ value[7,8]. Upon the fact that $IC_{50}$ values of NCV2SG53 are 10-fold lower than those of NCV2SG48 against Delta and some other variants, we tested an antibody cocktail consisting of NCV2SG48 and NCV2SG53. These antibodies in the cocktail did not obstruct each other and broadly neutralized all variants we tested (Fig. 2b). Collectively, potent bnAbs can be generated in convalescent patients infected by D614G early strain after 55 days from infection while most of the convalescent patients barely acquired bnAbs. We confirmed that the cocktail consisting of different monoclonal bnAbs is an effective way to achieve broadness of variant spectrum and effectiveness at a low dose that is attractive for the treatment of COVID-19 patients infected by SARS-CoV-2 variants and preventive administration to immunocompromised patients.

**Structural basis of monoclonal bnAbs**. To gain insight into how NCV2SG48 and NCV2SG53 neutralize broad variant spectrum, we determined the X-ray crystal structure of antigen-binding fragments (Fabs) in complex with the Wuhan S-RBD. From the structure analysis, we found that both antibodies differently recognize RBM close to each other to inhibit ACE2-binding (Fig. 3a). Neutralizing antibodies of SARS-CoV-2 can be classified based on binding region[8,24]. We found that NCV2SG48 is a typical class 1 antibody, while NCV2SG53 recognizes a class 2 epitope (Fig. 3b, c and Supplementary Fig. 4a). The heavy and light chains of both NCV2SG48 and NCV2SG53 interacted with RBM, which could compete with ACE2 for binding and block viral entry. The X-ray crystallography revealed that NCV2SG48 and NCV2SG53 form multiple hydrogen bonds with RBD directly or indirectly via water molecules (Fig. 3d, e). Notably, NCV2SG48 interacted with RBD by the extensive large interface, which covers almost the entire ACE2-binding region.

We next determined the binding interface area of neutralizing mAbs with Wuhan RBD. Compared to known neutralizing mAbs, heavy and light chains of NCV2SG48 had one of the largest

interaction areas among the mAbs we investigated (Fig. 4a). Interface areas of Fab from reported mAbs, NCV2SG48, and NCV2SG53 with Wuhan RBD were summarized (Fig. 4b). The binding interface area of NCV2SG48 (1200.4 Å$^2$) with Wuhan RBD accounts for 134–205% compared to that of reported class 1 REGN10933 (Casirivimab, 894.2 Å$^2$), class 3a REGN10987 (Imdevimab, 584.0 Å$^2$), and class 3b S309 (Sotrovimab, 763.0–747.5 Å$^2$) neutralizing mAbs (Supplementary Table 2). NCV2SG48 generated 40 hydrogen bonds with RBD including water-mediated bonds which possibly accept even bulky amino acids that emerged in Omicron BA.1 (Supplementary Fig. 5a). Because of differences in resolution and structural analysis methods, it is difficult to count the number of water molecule-mediated bonds in the previously reported neutralizing mAb-RBD complexes. However, even excluding the contribution of water molecules, the hydrogen bonds formed between NCV2SG48 and RBD were one of the most numerous (Supplementary Fig. 5b). Structure analysis suggests that SHM created five additional interaction sites which contribute to the extension of the binding interface with RBD covering the almost entire part of RBM including the conserved structure (Fig. 4c). Such an extremely large number of hydrogen bonds and binding interface well explain the stability of NCV2SG48 neutralizing potency to multiple mutations generated in variant RBDs (Fig. 4d and Supplementary Fig. 5c). To confirm the critical role of SHMs in the broadness of NCV2SG48 neutralization activity, we generated $V_H/V_L$ germline revertant and performed a neutralization assay. Germline reverted NCV2SG48 antibody substantially reduced neutralization activity to Alpha, Beta, Omicron BA.1, BA.2, BA.2.12.1, and BA.4/5 but the only minor reduction to Wuhan, D614G, and Delta variants indicating that the introduced SHMs play critical roles in acquiring broad neutralizing capacity against even unexposed variants (Figs. 2b, 4e). Thus, the acquired SHMs in the immunoglobulin gene of long-course COVID-19 patients were found to contribute significantly to the broadness of neutralizing antibodies.

To confirm these points, we further performed the X-ray crystal structure analysis of NCV2SG48 Fab in a complex with the Delta and Omicron S-RBD. The results showed that the binding angle between the RBD and Fab changed by BA.1 mutations, although the position of interaction was the same as that of the Wuhan RBD (Fig. 5a). The NCV2SG48 HC binding interface area was equivalent between Wuhan-Hu-1 and Omicron BA.1, while the LC binding interface area was reduced by about 22% (Fig. 5b). The overall Fab binding interface area was reduced by about 7%. The reduced binding interface area between NCV2SG48 LC and Omicron RBD could be explained by generated conformational conflict between NCV2SG48 and Omicron RBD at the mutated position, N501Y and Q498R, in Omicron RBD (Fig. 5a). The differences between the Cα atoms of each component of the RBD-NCV2SG48Fab complexes were less than 1 Å between Wuhan-Hu-1 and Omicron, indicating that their structures were highly conserved. These facts suggest that the steric hindrance caused by the bulky amino acid mutation of Omicron RBD (especially N501Y and Q498R) pushed up LC and weakened the RBD-Fab interaction (Fig. 5a). The binding interface of HC and LC has mostly conserved in Delta and Omicron BA.1 RBD, suggesting that the extended interaction area caused by the SHMs functioned effectively (Fig. 5b, c). The extension of RBD binding residues by NCV2SG48 accounts for 79.2% of conserved amino acid residues in Omicron BA.1 RBD (19 out of 24 residues) that presumably contributed to the resistance to heavy mutations (Fig. 5d).

It has been demonstrated that the use of mAbs targeting the S protein is a powerful way to treat COVID-19 patients, however, the emergence of antibody-resistant escape mutants remains a concern[25,26]. To evaluate the susceptibility of our neutralizing antibodies to the appearance of escape mutants, we cultured authentic SARS-CoV-2 virus in the presence of each mAb. We detected multiple escape mutants from the culture with

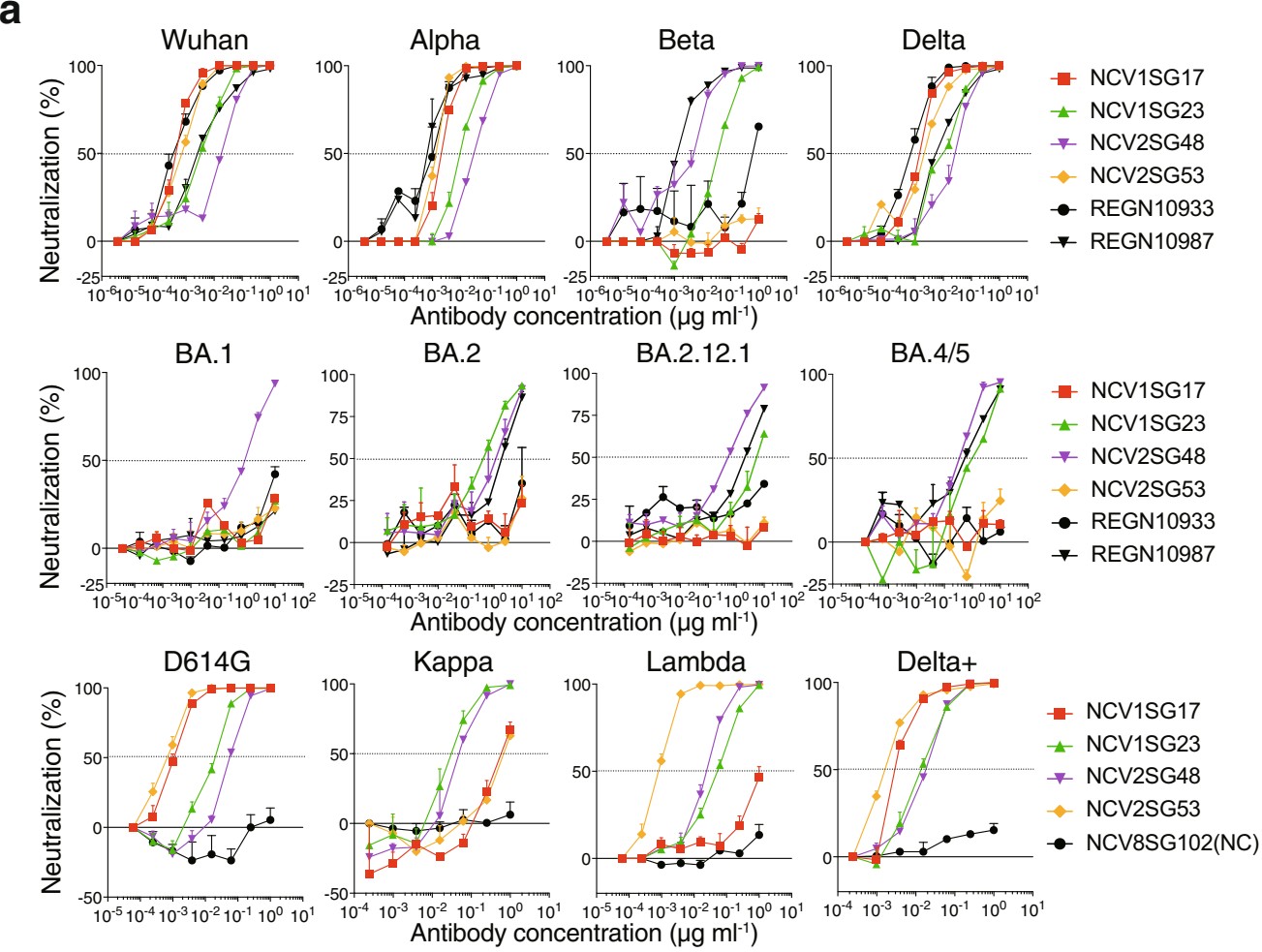

**Fig. 2 Broadly neutralizing potency of mAbs against SARS-CoV-2 variants. a** Dose-response analysis of the neutralization by each mAb NCV1SG17, NCV1SG23, NCV2SG48, NCV2SG53, REGN10933, and REGN10987 on indicated variants of SARS-CoV-2 pseudovirus. The horizontal dotted line on each graph indicates 50% neutralization. Data are mean ± SEM of technical replicates from 2 to 3 independent experiments. **b** The IC$_{50}$ neutralization values of each and the mixture of mAbs against indicated pseudoviruses of the SARS-CoV-2 variant. NCV2SG48-GL, germline revertant of NCV2SG48.

**b**

| Pseudovirus neutralization | IC$_{50}$ (ng ml$^{-1}$) | | | | | | | | | | | |
|---|---|---|---|---|---|---|---|---|---|---|---|---|
| | Wuhan | D614G | Alpha | Beta | Delta | Delta+ | Kappa | Lambda | BA.1 | BA.2 | BA.2.12.1 | BA.4/5 |
| NCV1SG17 | 0.5 | 1.1 | 2.4 | >1,000 | 1.5 | 3.8 | 500 | >1,000 | >10,000 | >10,000 | >10,000 | >10,000 |
| NCV1SG23 | 4.0 | 19 | 11 | 37 | 11 | 15 | 35 | 55 | >10,000 | 435 | 4,920 | 767 |
| NCV2SG48 | 21 | 59 | 34 | 11 | 27 | 19 | 52 | 26 | 798 | 1,028 | 497 | 382 |
| NCV2SG53 | 0.8 | 0.7 | 1.4 | >1,000 | 2.3 | 1.7 | 707 | 0.9 | >10,000 | >10,000 | >10,000 | >10,000 |
| NCV2SG48 +NCV2SG53 | 1.4 | 1.0 | 2.3 | 11 | 2.9 | 0.9 | 35 | 1.2 | 848 | 1,633 | 748 | 418 |
| REGN10933 | 0.5 | 0.7 | 1.0 | 299 | 0.8 | n.d. | n.d. | n.d. | >10,000 | >10,000 | >10,000 | >10,000 |
| REGN10987 | 2.9 | 0.6 | 0.7 | 1.4 | 6.8 | n.d. | n.d. | n.d. | >10,000 | 1,828 | 1,934 | 627 |
| REGN10933 +REGN10987 | 0.6 | 0.9 | 0.9 | 1.4 | 0.8 | 2.3 | 5.5 | 2.9 | >10,000 | 1,774 | 1,252 | 438 |
| NCV2SG48-GL | 119 | 143 | >10,000 | >10,000 | 54 | n.d. | n.d. | n.d. | >10,000 | >10,000 | >10,000 | >10,000 |

NCV1SG17 or NCV2SG53 and identified S494P mutation or E484D/G485D/G485R mutations, respectively. However, we detected none of the escape mutants from seven independent cultures under the presence of NCV1SG23 or NCV2SG48 alone suggesting that these two mAbs are highly resistant to spontaneous mutations (Supplementary Fig. 6).

NCV2SG53 also recognizes RBM at the class 2 site (Fig. 3a and Supplementary Fig. 4a). In contrast to NCV2SG48 which

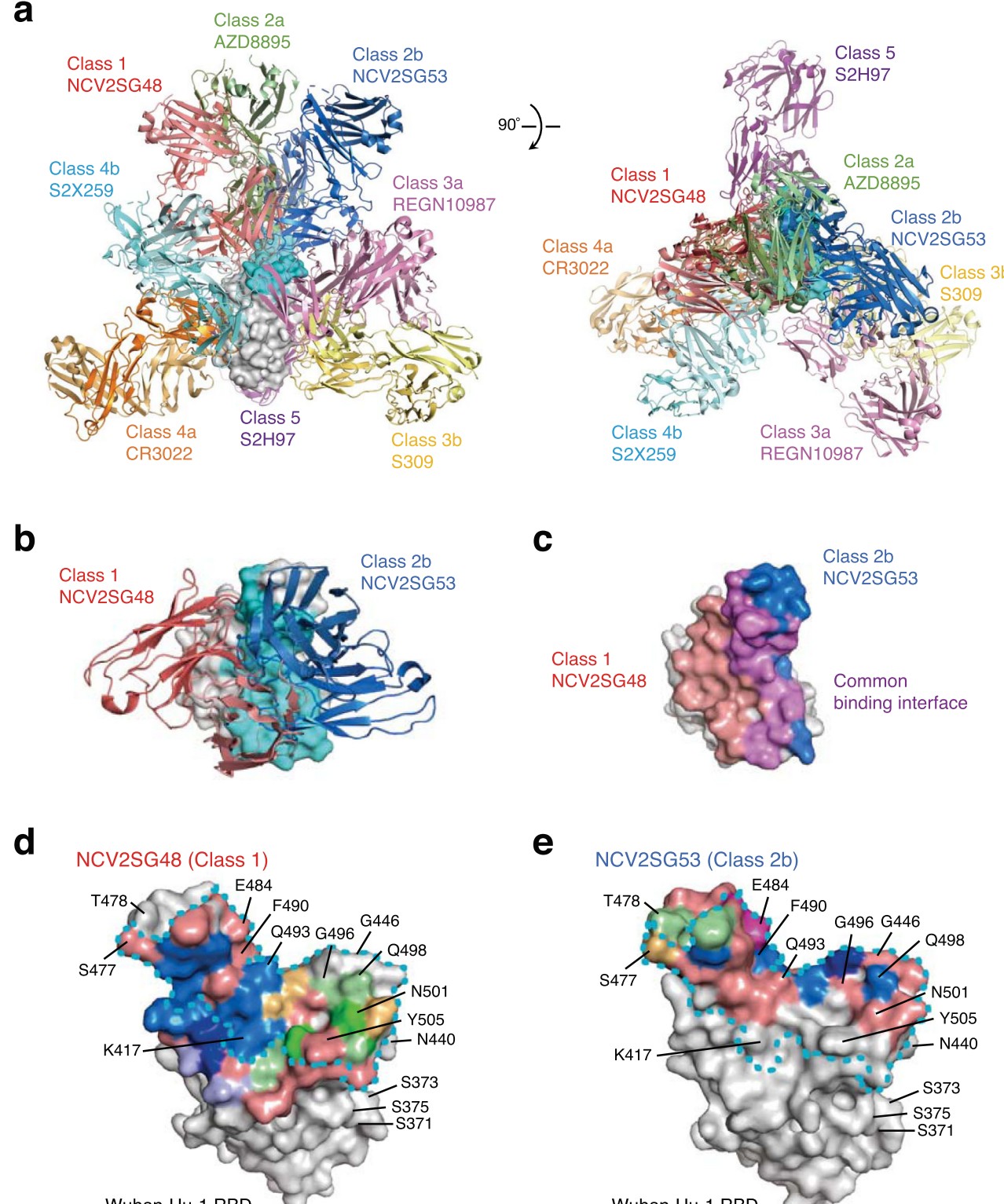

**Fig. 3 Structural epitope map of NCV2SG48 and NCV2SG53 binding to the SARS-CoV-2 RBD. a** The structural overview of Fab variable regions of NCV2SG48 (Class 1) and NCV2SG53 (Class 2b) together with a reported different class of neutralizing mAbs, AZD8895 (Class 2a), REGN10987 (Class 3a), S309 (Class 3b), CR3022 (Class 4a), S2X259 (Class 4b), and S2H97 (Class 5), complexed with Wuhan RBD. RBD is shown as the surface model (*gray*) with the ACE2-binding interface (*cyan*). **b** NCV2SG48 and NCV2SG53 complexed with Wuhan RBD. **c** Binding interface on Wuhan RBD with NCV2SG48 (*pink*), NCV2SG53 (*blue*), or common (*purple*). **d, e** The Binding interface (*pink*) of NCV2SG48 and NCV2SG53 are shown on the surface model of Wuhan RBD. The ACE2-binding site is surrounded by a dotted line (*cyan*). Residues forming a hydrogen bond with HC (*blue*), HC via water (*light purple*), HC and HC via water (*dark blue*), LC (*light green*), LC via water (*gold*), LC and LC via water (*green*), and HC, HC via water, and LC (*purple*) are indicated.

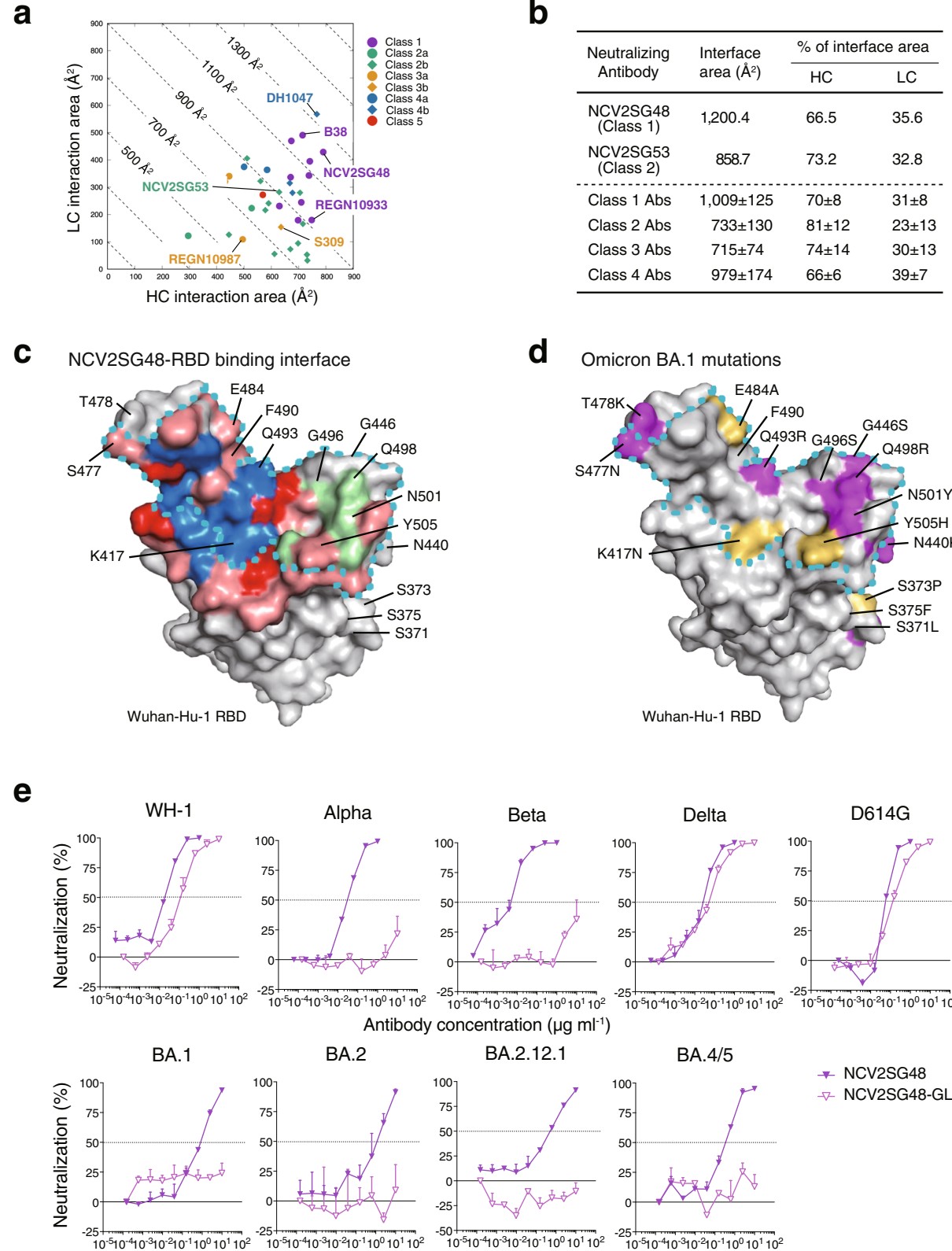

recognizes an extremely large interaction area, the binding of NCV2SG53 to RBD is highly dependent on interaction with E484 residue on RBD because 6 hydrogen bonds were formed around E484 residue (Fig. 6a). Structure analysis of NCV2SG53 with Wuhan and Delta RBDs suggested that SHM created two interaction sites which presumably contributed to the extension

of the binding interface and increased binding affinity to RBD (Fig. 6b and Supplementary Fig. 5c). T57S mutation in HC-CDR2 could contribute to reducing the steric hindrance with F490 of RBD and increasing affinity around E484 (Fig. 6a, b). This explained well why the E484 mutation emerged in Beta and Omicron abolished neutralization by NCV2SG53 (Fig. 2). The

**Fig. 4 SHMs contributed to additional hydrogen bonds for the extended binding interface between NCV2SG48 and RBD. a** Binding interface areas of the HC and LC from NCV2SG48, NCV2SG53, and reported class 1–5 neutralizing mAbs with RBD are plotted. **b** Fab-RBD interface areas from (**a**) with the percent contribution of HC and LC are shown by mean ± SD. **c** NCV2SG48-RBD binding interface on the RBD surface model. Residues forming a hydrogen bond with HC (*blue*) and LC (*green*) are indicated. Hydrogen bonds generated by SHM (*red*) are shown on a binding interface (*pink*). **d** Mutated residues of Omicron variant on RBD surface model. Mutation to bulkier side chains (*purple*) or to non-bulkier side chains (*yellow*) is shown. **c, d** The ACE2-binding site is surrounded by a dotted line (*cyan*). **e** Dose-response analysis of the neutralization by NCV2SG48 and its germline revertant (NCV2SG48-GL) on indicated variants of SARS-CoV-2 pseudovirus. The horizontal dotted line on each graph indicates 50% neutralization. Data are mean ± SEM of technical duplicates.

importance of E484 and neighboring amino acids is further confirmed by escape mutant analysis (Supplementary Fig. 6). On the other hand, mutations other than E484 such as Delta which has L452R/T478K mutations maintained a binding interface with NCV2SG53 and did not weaken the neutralization activity of NCV2SG53 (Figs. 2b, 6c, d). Collectively, NCV2SG53 can neutralize diversified variants that do not have E484 or G485 mutation while NCV2SG48 is highly resistant to broad mutations by forming an extremely large binding interface with RBD. In both cases, SHM introduced in the course of GC reaction is thought to play a critical role in universal inhibition against diversified SARS-CoV-2 variants.

## Discussion

The decrease of antibody efficacy of vaccinated and naturally infected by early SARS-CoV-2 strain explains the breakthrough infections and reinfections by Omicron variants, however, we realized that some convalescent people acquire bnAbs counteracting a broad SARS-CoV-2 variant. Therefore, we aimed to address how frequently bnAbs are produced in convalescent individuals infected by early strain and how those antibodies can exert neutralization to a broad spectrum of variants including heavily mutated Omicron. Here we found that convalescent individuals who elicited polyclonal antibodies to early SARS-CoV-2 strain, D614G (B.1.1), barely possessed antibodies inhibiting Omicron. Neutralization assay with sera against variants was useful to identify blood donors to isolate bnAbs in addition to donor criteria, over 17 days of hospitalization with severe illness. Based on the present single cell-derived antibody analysis, the frequency of bnAbs in circulating S-trimer-specific B cells of the D614G-lineage convalescent patients after 17 days from diagnosis was about 4% (4 out of 102 mAbs) while only 2% (NCV1SG23 and NCV2SG48) was effective in Omicron variants (Fig. 2 and Supplementary Fig. 2c). A prolonged hospitalization period to 55 days increased the frequency of bnAbs compared to over 17 days in which bnAbs showed SHMs suggesting that bnAbs are generated through the GC reaction which is generally observed in the lymphoid tissues after the viral infection.

In this study, SHMs introduced in CDRs contributed to the key structure of bnAbs and broad interaction with spike variants. Since GCs in the lymphoid organs play critical roles in the accumulation of SHMs and selection of higher affinity antibodies, the present data in addition to previous reports indicate that GC reaction after the SARS-CoV-2 infection is necessarily required for the generation of bnAbs[19,27–29]. Based on the fact that bnAbs were isolated from patients with long hospitalization of nearly 2 months, longer exposure to viral antigens over a month could give rise to the selective amino acid replacements in the CDR regions creating additional hydrogen bonds to the target viral protein. The epitope recognized by NCV2SG48 or NCV2SG53 overlaps with RBD residues involved in the binding to ACE2, hence classified as class 1 and class 2 antibodies. Previous neutralization studies on pseudovirus showed most of the established mAbs of class 1, 2, 3, and 4 RBD and the NTD mAbs lost neutralizing activity completely or partially against Omicron BA.1[9]. In this study, the potency of class 1 and class 2 RBD mAbs all dropped by >100-fold. In contrast, the potency of

NCV2SG48 class 1 antibody showed only a 13.5-fold reduction to Omicron BA.1 to D614G (Fig. 2b), indicating that targeting of conserved residues with numerous hydrogen bonds could lead to neutralization breadth and resilience to antigenic shift associated with viral evolution even in the RBM-specific class 1 mAb. Among the Emergency Use Authorization (EUA) mAbs, only class 3 mAbs such as AZD1061 and Vir-7831/S309, but none of class 1 and class 2 mAbs, showed a broad inhibition including Beta, Omicron BA.1, BA.2, BA.2.12.1, and BA.4/5 (Supplementary Fig. 4b). Therefore, the broadness of neutralizing activity of NCV2SG48 which targets RBM, not RBD stem, is a unique feature and it will be interesting to clarify how mAb effectively neutralizes viral variants.

The structural analysis revealed that NCV2SG48 and NCV2SG53 recognize partially overlapped epitopes (Fig. 3c and Supplementary Fig. 5c), however, the mAb cocktail consisting of NCV2SG48 and NCV2SG53 acts in a complementary manner. Namely, this class 1/2 cocktail can neutralize SARS-CoV-2 variants with a lower IC$_{50}$ value without intercepting the other. Although structural analysis of NCV2SG48 in complex with RBD revealed an extremely large interface almost entirely covering RBM that can conformationally block interaction with ACE2, the functional importance of a large interface in viral neutralization has not been completely clarified.

NCV2SG48 antibody was rearranged by IGHV3-53 and IGKV1-9 (Supplementary Fig. 2), which combination was not detected in the isolated mAbs other than NCV2SG48 suggesting bnAbs may be preferentially derived from minor responders. This could explain why most of the convalescent sera showed extremely reduced neutralizing titers to Omicron. Because significantly higher neutralization activity was detected only in donors who experienced long-term hospitalization, the accumulated SHMs in the antibody must play an important role in increasing neutralization activity and the generation of broadly neutralizing antibodies. To our surprise, we could isolate bnAbs from 87- and 93-year-old donors even though it is accepted that elderly people are at high risk for serious symptoms or death[30–33]. Therefore, it is interesting whether the ability to generate neutralizing antibodies correlates with clinical outcomes in elderly people.

Collectively, activation and recall of the broadly reacting rare B cells may confer sustainable protection against SARS-CoV-2 variants emerging one after another. Present results will contribute to future efforts to develop vaccines and therapies counteracting SARS-CoV-2 variants.

## Methods

**Convalescent and uninfected human blood samples**. Volunteers aged 23 to 93 with a history of convalescent COVID-19 were enrolled from April 2020 to January 2021. Blood samples were collected on the day or one day before discharging from the hospital after symptom resolution. Duration is the time between PCR positive and blood sample collection. All blood samples used in this study were collected before taking any SARS-CoV-2 vaccination. Uninfected healthy volunteers aged 36 to 62 who do not have severe immunological symptoms such as immunodeficiency, autoimmune, and allergic diseases were enrolled, and we confirmed uninfected/unvaccinated donors by their clinical history and ELISA titer. Detailed information on the cohort is in Supplementary Table 1. PBMCs and plasma samples were isolated by density gradient centrifugation with Ficoll-Paque PLUS (GE Healthcare) and stored at −80 °C until use. The study was approved by the Ethical Committee for Epidemiology of Hiroshima University (E-2011) for studies involving humans. Informed consent was obtained from all subjects involved in the study.

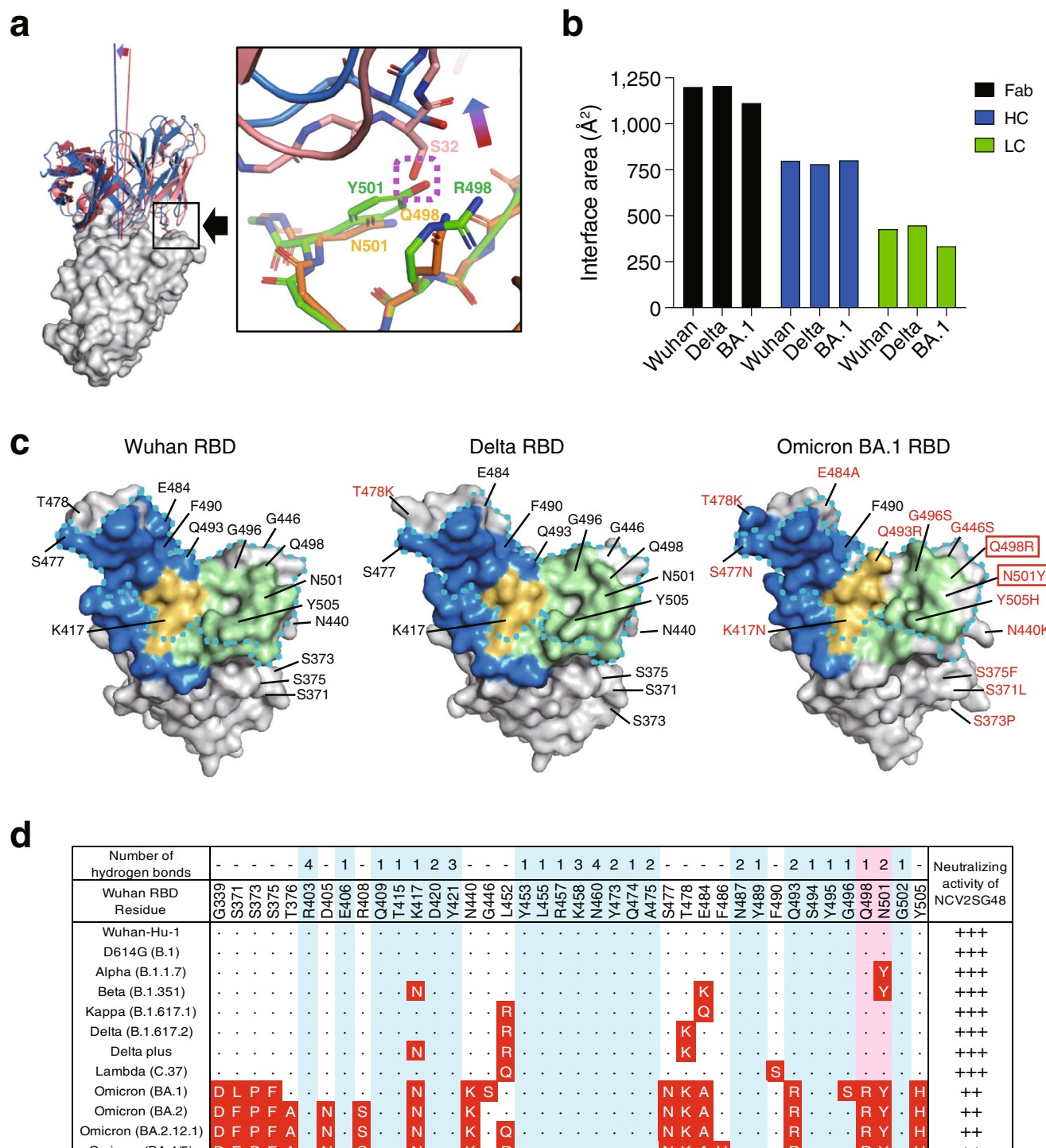

**Fig. 5 Conserved binding of NCV2SG48 on Omicron RBD. a** Fab of NCV2SG48 complexed with Wuhan RBD (*light red*) or Omicron RBD (*light blue*) is shown on the surface model of Wuhan-Hu-1 RBD. The binding angles of Fab to Wuhan RBD (*red line*) and Omicron RBD (*blue line*). The conformational conflict between NCV2SG48 and Omicron RBD is boxed and indicated by the arrow. Residues of a possible crush between NCV2SG48 and Omicron RBD are indicated by a purple dotted line. Wuhan-Hu-1 RBD (*orange*); Omicron RBD (*green*); S32 of NCV2SG48 LC, N501 and Q498 of Wuhan RBD, and Y501 and R498 of Omicron RBD are indicated. **b** The binding interface area of NCV2SG48 Fab, HC, and LC with Wuhan, Delta, and Omicron BA.1 RBDs. **c** Binding site of NCV2SG48 on the surface model of Wuhan RBD, Delta, and Omicron RBD. The binding interface of HC (*blue*), LC (*green*), or HC/LC overlapped (*yellow*) is shown. The ACE2-binding site is surrounded by a dotted line (*cyan*). Mutated amino acids are shown in red and residues causing conformational conflict with NCV2SG48 are boxed. **d** The matrix represents amino acid substitutions present in RBD of indicated SARS-CoV-2 variants. The substituted amino acids are shown in red boxes. The numbers of hydrogen bonds of NCV2SG48 to Wuhan RBD are indicated on the corresponding amino acid residue. Residues of blue interact with SARS-CoV-2 RBD. Residues of conformational conflict to Omicron RBD are highlighted in pink. IC$_{50}$ of neutralizing activity of NCV2SG48 to SARS-CoV-2 pseudoviruses is indicated (IC$_{50}$: +++, 10–99 ng/ml; ++, 100–999 ng/ml).

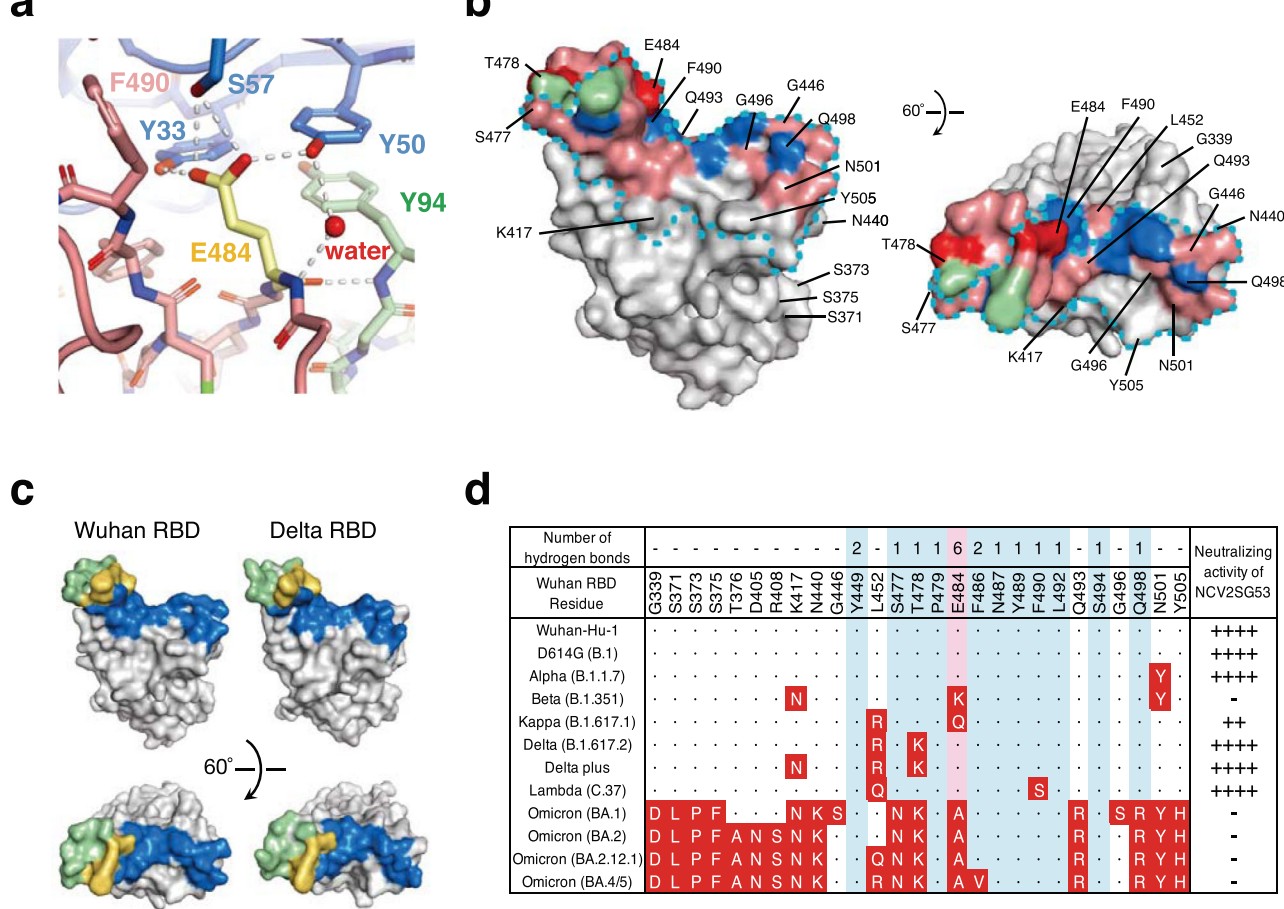

**Fig. 6 SHM contributed to hydrogen bonds between NCV2SG53 and E484 residue of RBD. a** Interaction of RBD E484 residue with NCV2SG53 mAb through the multiple hydrogen bonds. Among hydrogen bonds formed on RBD E484, S57 residue in the NCV2SG53 heavy chain CDR2 was generated by the SHM and contributes to increased binding affinity. **b** NCV2SG53-RBD binding interface on the Wuhan RBD surface model. Residues forming a hydrogen bond with HC (*blue*) and LC (*green*) are indicated. Hydrogen bonds generated by SHM (*red*) are shown on the binding interface (*pink*). The ACE2-binding site is surrounded by a dotted line (*cyan*). **c** Comparison of the NCV2SG53 binding interface between Wuhan RBD and Delta RBD. The binding interface of HC (*blue*), LC (*green*), or HC/LC overlapped (*yellow*) is shown on the surface model of RBD. **d** The matrix represents amino acid substitutions present in RBD of indicated SARS-CoV-2 variants. The substituted amino acids are shown in red boxes. The numbers of hydrogen bonds of NCV2SG53 to Wuhan RBD are indicated on the corresponding amino acid residue. Residues of blue interact with SARS-CoV-2 RBD. E484 in RBD is the most important for binding with NCV2SG53 (*pink*). Mutation in the E484 residue of SARS-CoV-2 RBD reduces the binding of NCV2SG53. IC$_{50}$ of neutralizing activity of NCV2SG53 to SARS-CoV-2 pseudoviruses is indicated (IC$_{50}$: ++++, <10 ng/ml; ++, 100–999 ng/ml; −, >1000 ng/ml).

**Flow cytometry for single-cell sorting**. For single-cell sorting, PBMCs were treated with FcX blocking antibodies (BioLegend, #4422302) to reduce non-specific labeling of the cells. PBMCs were stained with S-trimer-Strep-tag, CD19-APC-Cy7 (BioLegend, #302217), and IgD-FITC (BioLegend, #348206) for 20 min on ice. After washing, cells were stained with Strep-Tactin XT-DY649 (IBA) for 20 min on ice. The cells were resuspended in FACS buffer (PBS containing 1% FCS, 1 mM EDTA, and 0.05% NaN3) supplemented with 0.2 μg/ml propidium iodide (PI) to exclude dead cells. Cell sorting was performed on Special Order System BD FACSAria II (BD Biosciences) to isolate S-trimer+ CD19+ IgD− cells from the PI− live cell gate. Cells were directly sorted into a 96-well PCR plate. Plates containing single-cells were stored at −80 °C until proceeding to RT-PCR. Flow cytometric data were acquired on BD LSRFortessa (BD Biosciences) or CytoFLEX S (Beckman Coulter). Flow cytometric data were analyzed using BD FACSDiva (v8.0.2, BD Biosciences), CytExpert software (v2.4, Beckman Coulter), or FlowJo software (v10.8.1, BD Biosciences).

**Single-cell RT-PCR and monoclonal antibody production**. Single-cell sorted PCR plates were added to each well by 2 μl of pre-RT-PCR mix containing the custom reverse primers. After heating at 65 °C for 5 min, plates were immediately cooled on ice. 2 μl of the pre-RT-PCR2 (PrimeScript™ II Reverse Transcriptase, Takara Bio) mix was added to each well. For RT reaction, samples were incubated at 45 °C for 40 min followed by heating at 72 °C for 15 min, then cooled on ice. For PCR amplification of full-length immunoglobulin heavy and light chain genes, PrimeSTAR DNA polymerase (Takara Bio) and custom primers were used. For first PCR, the initial denaturation at 98 °C for 1 min was followed by 25 cycles of sequential reaction of 98 °C for 10 s, 55 °C for 5 s, and 72 °C for 1.5 min. For second

PCR, the initial denaturation at 98 °C for 1 min was followed by 35 cycles of sequential reaction of 98 °C for 10 s, 58 °C for 5 s, and 72 °C for 1.5 min. PCR fragments were assembled into a linearized pcDNA vector using NEBuilder HiFi DNA Assembly Master Mix (New England Biolabs) according to the manufacturer's instructions. The pcDNA3 (Invitrogen) vectors containing an Ig light chain gene and the pcDNA4 (Invitrogen) vectors containing an Ig heavy chain gene were simultaneously transfected into Expi293 cells using Expi293 Expression System Kit (Thermo Fisher Scientific). Four days after the transfection, the culture supernatants were collected and subjected to ELISA.

**Production of recombinant S-trimer and RBD**. For the production of recombinant S-trimer, soluble S protein (amino acids 1–1213), including the T4 foldon trimerization domain, a histidine tag, and a strep-tag, was cloned into the mammalian expression vector. The protein sequence was modified to remove the polybasic cleavage site (RRAR to A), and two stabilizing mutations were also introduced (K986P and V987P; wild-type numbering)[34,35]. The human codon-optimized nucleotide sequence encoding for the S protein of SARS-CoV-2 (GenBank: MN994467) was synthesized commercially (Eurofins Genomics). A soluble version of the S protein (amino acids 1–1213), including the T4 foldon trimerization domain, a histidine tag, and a strep-tag, was cloned into the mammalian expression vector pCMV. The protein sequence was modified to remove the polybasic cleavage site (RRAR to A), and two stabilizing mutations were also introduced (K986P and V987P; wild-type numbering)[35]. The gene encoding RBD of SARS-CoV-2, Wuhan-Hu-1, was synthesized and cloned into vector pcDNA containing a human Ig leader sequence and C-terminal 6xHis tag. RBD mutants

were generated by overlap PCR using primers containing mutations. The vector was transfected into Expi293 cells and incubated at 37 °C for 4 days. Supernatants were purified using Capturem™ His-Tagged Purification kit (Takara Bio), then dialyzed by PBS buffer overnight. Protein purity was confirmed by SDS-PAGE. Protein concentration was determined spectrophotometrically at 280 nm.

**ELISA for SARS-CoV-2 antigens.** MaxiSorp ELISA plates (Thermo Fisher Scientific) were coated with 2 µg/ml purified spike RBD or trimer in 1xBBS (140 mM NaCl, 172 mM H3BO3, 28 mM NaOH) overnight at 4 °C, and then blocked with blocking buffer containing 1% BSA in PBS for 1 h. Antibodies diluted in Reagent Diluent (0.1% BSA, 0.05% Tween in Tris-buffered Saline) were added and incubated for 2 h. HRP-conjugated antibodies were added and incubated for 2 h. Wells were reacted with the TMB substrate (KPL) and the reaction was stopped using 1 M HCl. The absorbance at 450 nm was measured on iMark Microplate Reader (Bio-Rad) and analyzed on MPM 6 software (Bio-Rad). Antigen-specific Ig titers were determined using serial serum dilution on antigen-coated wells next to Ig-capturing antibody standard wells on the same ELISA plate.

**Affinity measurement using biolayer interferometry (BLI).** The binding affinity of obtained antibodies to RBD was examined by the BLItz system (Sartorius Japan) using protein A-coated biosensors. 10 µg/ml of antibody was captured by the biosensor and equilibrated, followed by sequential binding of each concentration of RBD. For dissociation, biosensors were dipped in PBST for 900 sec. Results were analyzed on BLItz Pro (v1.3.1.3, Molecular Devices).

**Reagents.** psPAX2 (Addgene, no.12260) was a gift from Didier Trono. pCDNA3.3_CoV2_B.1.1.7 (Addgene, no.170451) for Alpha-S and pcDNA3.3-SARS2-B.1.617.2 (Addgene, no.172320) for Delta-S proteins, were gifts from David Nemazee[36]. pTwist-SARS-CoV-2 Δ18 B.1.351v1 (Addgene, no.169462) for Beta-S protein was a gift from Alejandro Balazs[37]. Lentiviral vector, pWPI-ffLuc-P2A-EGFP for luciferase reporter assay and pTRC2puro-ACE2-P2A-TMPRSS2 for the generation of 293T cell line susceptible to SARS-CoV-2 infection was created from pWPI-IRES-Puro-Ak-ACE2-TMPRSS2, a gift from Sonja Best (Addgene, no.154987) by In-Fusion® technology (Takara Bio). pcDNA3.4 expression plasmids encoding SARS-CoV-2 S proteins with human codon optimization and 19 a.a deletion of C-terminus (C-del19) from Wuhan, D614G, and Omicron were generated by assembly of PCR products, annealed oligonucleotides, or artificial synthetic gene fragments (Integrated DNA Technologies, IDT) using In-Fusion® technology. For Delta plus, Kappa and Lambda variants, S proteins with only RBD, D614, and P681 mutations were created from pcDNA3.4 encoding human codon-optimized Wuhan S protein (C-del19). LentiX-293T cells (Takara Bio) and 293T cells were maintained in culture with Dulbecco's Modified Eagle's Medium (DMEM) containing 10% fetal bovine serum (FBS), penicillin-streptomycin (Nacalai tesque), and 25 mM HEPES (Nacalai tesque).

**Generation of 293T cells stably expressing human ACE2 and TMPRSS2.** To generate stable 293T-ACE2.TMPRSS2 cells (293T/TRCAT), lentiviral vector VSV-G-pseudotyped lentivirus carrying ACE2 and TMPRSS2 genes were produced in LentiX-293T cells (Clontech) by transfecting with pTRC2puro-ACE2-P2A-TMPRSS2, psPAX2 (gag-pol), and pMD2G-VSV-G (envelope) using PEI-MAX (Polysciences). Packaged lentivirus was used to transduce 293T cells (Applied Biological Materials) in the presence of 5 µg/mL polybrene. At 72 h post-infection, the resulting bulk transduced population positive for Human ACE2 expression stained by FITC-anti-ACE2 Antibody (Sinobiological) was sorted by Special Order System BD FACSAria II (BD Biosciences) and maintained in the culture medium in the presence of 2 µg/ml of puromycin.

**Pseudovirus production and neutralization.** Pseudoviruses bearing SARS-Cov2 S-glycoprotein and carrying a firefly luciferase (ffLuc) reporter gene were produced in LentiX-293T cells by transfecting with pWPI-ffLuc-P2A-EGFP, psPAX2, and either of S variant from Wuhan, D614G, Alpha, Beta, Delta, Delta plus, Kappa, Lambda, or Omicron using PEI-MAX (Polyscience). Pseudovirus supernatants were collected approximately 72 h post-transfection and used immediately or stored at −80 °C. Pseudovirus titers were measured by infecting 293T/TRCAT cells for 72 h before measuring luciferase activity (ONE-Glo™ Luciferase Assay System, Promega, Madison, WI). Pseudovirus titers were expressed as relative luminescence units per milliliter of pseudovirus supernatants (RLU/ml). For neutralization assay, pseudoviruses with titers of 1–4 × 10^6 RLU/ml were incubated with antibodies or sera for 0.5 h at 37 °C. Pseudovirus and antibody mixtures (50 µl) were then inoculated with 5 µg/ml of polybrene onto 96-well plates that were seeded with 50 µl of 1 × 10^4 293 T/TRCAT cells/well one day before infection. Pseudovirus infectivity was scored 72 h later for luciferase activity measured on ARVO X13 and 2030 Workstation (Perkin Elmer). The serum dilution or antibody concentration causing a 50% reduction of RLU compared to control (ED50 or IC50, respectively) were reported as the neutralizing antibody titers. ED50 or IC50 were calculated using a nonlinear regression curve fit on Prism (v9.0, GraphPad).

**Neutralization assay with authentic SARS-CoV-2 viruses.** VeroE6/TMPRSS2 cells (African green monkey kidney-derived cells expressing human TMPRSS2, purchased from the Japanese Collection of Research Bioresources (JCRB) Cell Bank, JCRB1819, were maintained in DMEM containing 10% FBS and 1 mg/ml G418 at 37 °C in 5% CO2. The virus was propagated in VeroE6/TMPRSS2 cells and the virus titer was determined by the 50% tissue culture infectious dose (TCID50) method and expressed as TCID50/ml[38]. The viral strains used are SARS-CoV-2/JP/Hiroshima-46059T/2020 (B.1.1, D614G, EPI_ISL_6289932[39]), SARS-CoV-2/JP/HiroC77/2021, (AY.29, Delta, EPI_ISL_6316561), and SARS-CoV-2/JP/FH-229/2021 (BA.1.1, Omicron, EPI_ISL_11505197).

The serially diluted antibody (50 µl) was mixed with 100 TCID50/50 µl of the virus and reacted at 37 °C for 1 h, then inoculated into VeroE6/TMPRSS2 cells to determine the minimum inhibitory concentration (MIC). Alternatively, the infectivity of the reacted antibody-virus mixture was measured by inoculating to 8 wells of a 96-well plate and observing cytopathic effects (CPE) or by the plaque assay using 10% methylcellulose to determine 50% effective dose (ED50) of the antibody. SARS-CoV-2 infection was performed in the BSL3 facility of Hiroshima University.

**Escaping SARS-CoV-2 mutants from antibodies.** SARS-CoV-2 was incubated with mAb at twice the concentration of the EC50 corresponding to that viral load at 37 °C for 60 min. After the incubation, 100 µl of the mixture was added to one well of a 24-well plate with confluent VeroE6/TMPRSS2 cells and incubated for 72 h at 37 °C with 5% CO2. The supernatants were collected as an escape-mutant virus when CPE was manifested. A no-antibody-control included to confirm the amount of test virus required.

**Virus RNA Sequencing.** Viral RNA was extracted from virus-infected culture medium by using Maxwell RSC Instrument (Promega, AS4500). cDNA preparation and amplification were done in accordance with protocols published by the ARTIC network (https://artic.network/ncov-2019) using V4 version of the ARTIC primer set from Integrated DNA Technologies to create tiled amplicons across the virus genome. The sequencing library was prepared using the NEB Next Ultra II DNA Library Prep Kit for Illumina (New England Biolabs, E7645). Paired-end, 300 bp sequencing was performed using MiSeq (Illumina) with the MiSeq reagent kit v3 (Illumina, MS-102-3003). Consensus sequences were obtained by using the DRAGEN COVID lineage software (Illumina, ver. 3.5.6). Variant calling and annotation were performed using the Nextclade website (https://clades.nextstrain.org).

**Preparation of RBD and Fab complexes.** RBD from Wuhan-Hu-1, Delta, and Omicron variant and Fab fragment from NCV2SG48 with 6xHis-tag expressed in Expi293F cells (Thermo Fisher Scientific) were purified using Ni-NTA Agarose resin (QIAGEN). Fab fragment of NCV2SG53 was isolated from papain digests of the monoclonal antibody expressed in Expi293F cells using HP Protein G column (Cytiva). Purified each Fab fragments and RBD were mixed in the molar ration of 1:1.2 and incubated on ice for 1 h. The mixture was loaded onto a Superdex 200 increase 10/300 GL column (Cytiva) equilibrated in 20 mM Tris-HCl pH7.5, 150 mM NaCl for removing the excess RBD. Fractions containing RBD and each Fab were collected and concentrated for crystallization. Chromatography was performed using NGC Chromatography Systems (BIO-RAD) and ChromLab v6 (BIO-RAD).

**Crystallization, X-ray diffraction data collection, and processing.** Crystallization was carried out by the sitting-drop vapor diffusion method at 20 °C. Crystals of RBD (Wuhan-Hu-1)-Fab (NCV2SG48) were grown in 2 µl drops containing a 1:1 (v/v) mixture of 7.5 mg/ml RBD solution and 0.1 M Bis-Tris pH5.5, 0.5 M ammonium sulfate and 19% PEG3350. Crystals of RBD (Delta)-Fab (NCV2SG48) were grown in 2 µl drops containing a 1:1 (v/v) mixture of 7.5 mg/ml RBD solution and 0.1 M Bis-Tris pH6.5, 17.5% PEG10000, 100 mM Ammonium acetate and 5% Glycerol. Crystals of RBD (Omicron BA.1)-Fab (NCV2SG48) were grown in 2 µl drops containing a 1:1 (v/v) mixture of 7.5 mg/ml RBD solution and 0.1 M Bis-Tris pH 5.5, 0.5 M ammonium sulfate, 19.5% PEG 3350, 1 mM EDTA and 10% glycerol. Crystals of RBD (Wuhan-Hu-1)-Fab (NCV2SG53) were grown in 2 µl drops containing a 1:1 (v/v) mixture of 7.5 mg/ml RBD solution and 0.1 M MES pH 6.0, 0.25 M ammonium sulfate and 22.5% PEG3350. Crystals of RBD (Delta)-Fab (NCV2SG53) were grown in 0.6 µl drops containing a 1:1 (v/v) mixture of 7.5 mg/ml RBD solution and 25% PEG1500. The single crystals suitable for X-ray experiments were obtained in a few weeks. X-ray diffraction data collections were performed using synchrotron radiation at SPring-8 beamline BL44XU[40] in a nitrogen vapor stream at 100 K. The data sets were indexed and integrated using the XDS package[41], scaled, and merged using the program *Aimless*[42] in the CCP4 program package[43]. The scaling statistics were shown in Table 1.

**Structure determination and analyses.** Phase determinations were carried out by the molecular replacement method using the program Phaser[44] in the PHENIX package[45] and the program Molrep[46] in the CCP4 program package with the combination of RBD structure (PDB ID:7EAM) and Fab structures (PDB ID:7CHB and 7CHP) as search models. The structure refinement was performed using the program phenix.refine[47] in the PHENIX package and the program coot[48] in the CCP4 program package. The final refinement statistics were shown in Table 1. Interactions between RBD and Fabs were analyzed using the program PISA[49] in

**Table 1 Data collection and refinement statistics Values in parentheses indicate the highest-resolution shell.**

| Antibody | NCV2SG48 | | | NCV2SG53 | |
|---|---|---|---|---|---|
| SARS-CoV-2 strain | Wuhan-Hu-1 | Delta | Omicron | Wuhan-Hu-1 | Delta |
| **Scaling statistics** | | | | | |
| Space group | $P2_122_1$ | $P2_12_12_1$ | $P2_1$ | $P2_12_12_1$ | $P6_122$ |
| Unit cell parameters (Å, °) | $a = 87.30$, $b = 102.94$, $c = 114.34$ | $a = 88.81$, $b = 113.73$, $c = 272.24$ | $a = 93.35$, $b = 115.68$, $c = 99.49$, $b = 105.89$ | $a = 76.26$, $b = 118.19$, $c = 196.29$ | $a = b = 77.61$, $c = 519.15$ |
| Resolution range (Å) | 102.94–2.18 (2.24–2.18) | 99.9–2.38 (2.42–2.38) | 49.50–3.30 (3.48–3.30) | 46.71–2.35 (2.40–2.35) | 67.21–3.05 (3.47–3.05) |
| Total number of reflections | 277,459 (22,507) | 1,520,712 (78,649) | 100,426 (15,956) | 371,488 (22,516) | 132,324 (5588) |
| Number of unique reflections | 54,347 (4404) | 111,340 (78,649) | 30,152 (4430) | 74,372 (4462) | 7419 (371) |
| Multiplicity | 5.1 (5.1) | 13.7 (14.4) | 3.3 (3.6) | 5.0 (5.0) | 17.8 (15.1) |
| $<I>/<\sigma (I)>$ | 5.1 (1.4) | 10.3 (1.5) | 5.5 (1.4) | 4.1 (1.0) | 8.9 (2.0) |
| CC1/2 | 0.984 (0.316) | 0.997 (0.770) | 0.995 (0.790) | 0.982 (0.685) | 0.998 (0.73) |
| Completeness (%) | 99.8 (99.9) | 100.0 (100.0) | 97.8 (99.0) | 99.6 (98.1) | 88.6 (61.9)[c] |
| **Refinement statistics** | | | | | |
| Resolution range (Å) | 43.65–2.18 | 42.98–2.38 | 47.84–3.30 | 46.71–2.35 | 67.21–3.06 |
| Number of reflections | 53,717 | 111,174 | 30,099 | 74,091 | 7,405 |
| $R_{work}$[a] (%)/$R_{free}$[b] (%) | 24.5/28.1 | 18.8/23.0 | 20.9/25.4 | 22.5/27.6 | 22.8/37.3 |
| Root-mean-square deviations | | | | | |
| Bond lengths (Å)/ angles (°) | 0.008/0.966 | 0.008/0.982 | 0.008/1.013 | 0.008/0.995 | 0.009/1.135 |
| Average B-factors (Å$^2$) /Number of atoms | | | | | |
| Protein (Chain R, H, L) | 46.5, 39.9, 39.76/1588, 1594, 1630 | 58.2, 51.3, 54.9/1563, 1594, 1630 | 109.2, 100.2, 101.7 /1571, 1636, 1581 | 61.4, 50.8, 47.8 /1552, 1627, 1644 | 75.44, 61.323, 74.16 /1522, 1639, 1619 |
| Protein (Chain A, B, C) | – | 71.5, 59.3, 57.5/1563, 1584, 1630 | 100.9, 102.1/, 108.1 /1636, 1590, 1576 | 62.7, 54.0, 4 7.3 /1561, 1671, 1625 | – |
| Protein (Chain D, E, F) | – | 64.0, 54.5, 51.36/1555, 1586, 1630 | – | – | – |
| Small molecules | – | – | 154.5/60 | 91.5/63 | – |
| Water molecules | 41.3 / 173 | 53.26/296 | – | 49.7/244 | – |
| Ramachandran plot | | | | | |
| Favored region (%) | 96.59 | 95.75 | 91.28 | 94.24 | 86.16 |
| Allowed region (%) | 3.25 | 3.76 | 8.64 | 5.20 | 13.68 |
| Outliers (%) | 0.16 | 0.49 | 0.08 | 0.56 | 0.16 |
| **PDB ID** | 7WNB | 8I5H | 7YOW | 7WN2 | 8I5I |

[a]$R_{work} = \sum|F_{obs} - F_{cal}|/\sum F_{obs}$, where $F_{obs}$ and $F_{cal}$ are the observed and calculated structure-factor amplitudes.
[b]The $R_{free}$ value was calculated using only an unrefined, randomly chosen subset of reflection data that were excluded from refinement.
[c]The completeness value was ellipsoidally calculated by the program *autoPROC*.

the CCP4 program package. All figures of structures were generated by the program pymol (The PyMOL Molecular Graphics System, Version 2.4.0., Schrödinger, LLC.). Class 2a/AZD8895 (PDB ID:7L7D), Class 3a/REGN10987 (PDB ID:6XDG), Class 3b/S309 (PDB ID:7JX3), Class 4a/CR3022 (PDB ID:6ZLR), Class 4b/S2X259 (PDB ID:7M7W), and Class 5/S2H97 (PDB ID:7M7W) open data were used in Fig. 3a.

**Statistics and reproducibility**. The statistical analysis was performed using Prism 9.0 (GraphPad, La Jolla, CA, USA). Ordinary One-way ANOVA, Two-way ANOVA, Kruskal-Wallis test, Wilcoxon rank test, and Friedman test were used to compare data. P-value 0.05 was considered statistically significant. Statistical tests are reported in figure legends and significance is reported at $p \leq 0.05$. To verify reproducibility, we repeated experiments more than two times as indicated in Figure legends. Detailed information on the sample is provided in Supplementary Tables.

**Reporting summary**. Further information on research design is available in the Nature Portfolio Reporting Summary linked to this article.

## Data availability
The structure of SARS-CoV-2 RBD in complex with NCV2SG48 and NCV2SG53 Fabs has been deposited in the Protein Data Bank. PDB ID: NCV2SG48-Wuhan RBD, 7WNB; NCV2SG48-Delta RBD, 8I5H; NCV2SG48-BA.1 RBD, 7YOW; NCV2SG53-Wuhan RBD, 7WN2; NCV2SG53-Delta RBD, 8I5I. Source data for figures are provided in the paper in Supplementary Data 1–3 or available from the corresponding author upon request. Monoclonal antibodies will be available upon request for research use. Human subjects will be available for researchers who are not listed in the human study protocol.

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

## Acknowledgements

We thank all study participants for our research; Blood donors, medical staff of the Hiroshima University Hospital, Shobara Redcross Hospital, Hiroshima Prefectural Hospital. We thank T. Utsumi, S. Eto, Y. Tamura, T. Kawahara for the arrangement of donors and collecting blood; T. Kawaguchi, N. Tani, and Y. Hayashi for technical assistance; N. Kikkawa for administrative assistance. All lab members for useful discussion and comments. We also thank the staff of the Analysis Center of Life Science, Hiroshima University for the use of their facilities. This work was performed using a synchrotron beamline BL44XU at SPring-8 (Harima, Japan) under the Collaborative Research Program of the Institute for Protein Research, Osaka University (Proposal No. 2021B6651, 2022A6728, 2022B6728). This work was supported by the JSPS KAKENHI Grant Numbers JP17H06937, JP18H02669, JP19K22538, and JP21H02751 to T.Y.; Sumitomo Mitsui Trust Bank-New Corona Vaccine and Therapeutics Development Donation Account to T.S. and T.Y.; Japan Agency for Medical Research and Development (AMED) Research Grant for COVID-19, JP20fk0108453 to T.S. and T.Y; Hiroshima Prefecture-Hiroshima University Government-Academia Collaboration COVID-19 Research Fund to T.S.; AMED Practical Research Project for Rare/Intractable Diseases, JP20fk0108531 to S.O.; AMED Translational Research Program, PreB 20334760 to N.N. and Y.Y.; Ministry of Education, Culture, Sports, Science and Technology grant 20H05773 to T.H.; and JST CREST Grant Number JPMJCR20H8 to T.H.

## Author contributions

K.S., Y.Ka. Y.M. and T.Y. designed the experiments. T.T. provided blood from patients. K.S., A.H., A.Y., T.H., R.Y., and H.A. produced and purified recombinant proteins. K.S. and N.N. sequenced and produced antibodies. K.S., N.N. and S.H. performed protein binding assays. Y.Ka. and A.I. performed pseudovirus neutralization assays. T.S., M.K., and Y.Ki performed neutralization and escape assays with authentic SARS-CoV-2. A.H. and A.Y. performed X-ray crystallography and analyzed the structure data. S.Oh. performed bioinformatic analysis. T.H. provided key reagents. K.S., Y.Ka. and T.Y. prepared the paper with input from all authors. Y.Y., S.Ok, T.S., and T.Y. supervised the study.

## Competing interests

The authors declare no competing interests.
