## [Peer Review File · Communications Biology]

Reviewers' comments:

Reviewer #1 (Remarks to the Author):

In this manuscript Shitaoka et al., used sera from SARS CoV-2 infected individuals to identify broadly neutralizing antibodies (bnAbs). They showed that individuals needing oxygenation treatment had the highest antibody titers and highest serum neutralization activity, and that two of these patients showed Omicron pseudovirus neutralization activity. The authors sorted PBMCs from five donors that were hospitalized for more than 17 days and processed the B cells to produce mAbs. They proceeded to run a nice analysis of the V gene usage and amount of somatic hypermutation (SHM) of these isolated mAbs showing that bnAbs might arise from rarer combinations of heavy and light chain V gene combinations. Two of these sorted mAbs, NCV2SG48 and NCV2SG53, were able to neutralize pseudovirus of many variants including two that showed neutralization activity against Omicron. The authors structurally characterized the Fabs of these two mAbs bound to wuhan-Hu-1 RBD and NCV2SG48 to Omicron RBD. They provide detailed analysis as to why these antibodies provide breadth.

Overall, this is good study that provides novel information about bnAbs against SARS CoV-2 that could be useful for guiding future vaccine research or antibody-based treatments. However, it lacks several experiments that are expected in this crowded research field. With major revisions, I would recommend this manuscript for publication.

Major revisions:

- 1) Authentic virus neutralization assays are expected in this field. While the authors performed these assays with D614G and a single assay of Delta, this should be expanded to all the variants tested in the pseudovirus assay. At the very least, the delta neutralization assay should be repeated, and omicron needs to be included.
- 2) The authors claim that SHM created more interactions between NCV2SG48 and the RBD. This could be tested by making germline reversion of the V-genes and running binding or neutralization assays. It would be useful to have an antibody sequence showing the sites of SHM and binding residues.
- 3) In Fig. 5a, determining the binding angle using the entire Fab is not appropriate. The Fv and Fc are connected by a flexible hinge and the conformation of the Fc in relation to the Fv in a crystal is likely more dependent on crystal contacts than biological relevant influences. This analysis should be limited to the angle of the Fv region. Also, please indicate how center of gravity was determined.
- 4) In the method for the convalescent or vaccinated human donors section, please clarify what the healthy volunteers were used for and the timing of the blood samples taken after vaccination. Are these the "uninfected" donors used in the study? I find it confusing if these are the uninfected controls, how do they show S specific IgM, IgG, and IgA levels? How were they confirmed to be uninfected?
- 5) All figures should include definitions of what error bars or box plots represent and the number of replicates. Fig. 1 and Extended Data Fig. 1 are missing this information.

Minor Revisions:

- 1) The manuscript would benefit from some clarifying edits in addition to point 4 above. For example, the first two sentences of the introduction are redundant as they are restating nearly the same thing. The sentence starting with "Notably..." on page 8, line 9 was also confusing and required multiple reads to understand what was being described.
- 2) The Rfree test set for the NCV2SG28 Fab-wuhan-Hu-1 structure is only 3.7% of reflections instead of the typical 5%. Why is this? The other two structures both use 4.9% of reflections.
- 3) In Extended Data Fig. 2e-f, the column for the CDR3 sequences needs to be adjusted so that the letters are legible
- 4) In the crystallization section of the methods, the units for the protein concentration should read mg/mL or mg mL⁻¹ with the -1 superscripted.
- 5) Include PDB ID for the omicron structure in the Data and availability section.

Reviewer #2 (Remarks to the Author):

Shitaoka et al. analyzed neutralizing antibodies isolated from long-term hospitalized convalescent patients infected by the SARS-CoV-2, including the broadly neutralizing antibody NCV2SG48. Furthermore, the complex crystal structures were determined to reveal the mode of action of two broadly neutralizing antibodies. NCV2SG48 binds RBD with a large binding interface and extensive interactions with conserved residues required for binding ACE2. Combination of two different neutralizing antibodies confer better antiviral effect and can neutralizing all the strains tested.

1. There were similar broadly neutralizing antibodies reported. The author should have a systematic comparison to these reported broadly neutralizing antibodies.
2. Many broadly neutralizing antibodies showed neutralizing activity against BA.1, however, lost their neutralizing activity against BA.4 and BA.5. Could the author do additional assays on BA.4 or BA.5? or based on the complex structure, have a prediction on the neutralizing potential of NCV2SG48 against BA.4 or BA.5. Will the binding of NCV2SG48 be affected by the mutations of BA.4 or BA.5?
3. An interesting point raised by the research is that the percentage of broadly neutralizing antibodies increases as the immune system has a longer interaction with the virus, which was supposed to be the results of accumulated somatic hypermutations. This could be very helpful in guiding the development of vaccines. However, the number of the samples is not big enough to make such a conclusion. Has this been observed by others? Or would data from similar research support this?

Reviewer #3 (Remarks to the Author):

This paper reports one antibody named NCV2SG48 isolated from patients with long hospitalization, which confers potent and broad neutralization against SARS-CoV-2 variants including Omicron. Furthermore, the authors determined the crystal structures of complexes of NCV2SG48 and RBD (both prototype and Omicron), and found that somatic mutations introduced in CDRs contributed to an extended binding interface, as well as the contact between hydrogen bonds and conserved residues at the region of RBM based on the structural analysis. Overall, the work is of value and may contribute to the development of broad-spectrum antibodies. However, there are some points that should be addressed before the work can be considered for publication.

Major Notes

1. Figure 1 and figure 2: The comparison should include other Omicron subvariants including BA.2, BA.2.12.1, and BA.4/BA.5 .
2. Line 20 page 6: Only prototype RBD was used in the binding affinity measurement of antibodies, which could not support in estimating the changes of affinity among different strains. Thus, the test on other Omicron subvariants would be helpful, see the above comment.
3. Figure 2b: The color scheme for the two curves of antibodies combination should be improve to distinguish them in far different color.
4. Line 24 page 7: According to the previous report, antibodies cocktails contain two or more non-overlapping antibodies to enhance the neutralizing potency and decrease the potential for SARS-CoV-2 escape mutants. But the NCV2SG48 and NCV2SG53 recognize the partially overlapping epitopes, causing theoretically competitive binding to RBD between NCV2SG48 and NCV2SG53. So, the author should explain the possible reason for still highly neutralizing effectiveness at a low dose when using NCV2SG48 and NCV2SG53 cocktails? Are they synergetic in function?
5. Figure 3: The two complex structures should be shown separately. You may add them to some appropriate place in Fig. 3.
6. The colors rendering of structural figures are confusing. The authors should keep consistent colors for models and footprints. And in the footprints, it should be revised to discriminate with similar colors as models.

Point-by-point responses (MS: COMMSBIO-22-2894-T)

We greatly appreciate all reviewers for their comments and suggestions that helped us to substantially improve our manuscript. According to the given comments, we carefully revised the manuscript and prepared responses on a point-by-point basis as follows. Changes from the original manuscript were indicated by red letters in the revised manuscript. We hope that our responses are satisfactory and that the revised manuscript will be acceptable for publication by *Communications Biology*.

Reviewer 1*Major revisions:*

1) *Authentic virus neutralization assays are expected in this field. While the authors performed these assays with D614G and a single assay of Delta, this should be expanded to all the variants tested in the pseudovirus assay. At the very least, the delta neutralization assay should be repeated, and omicron needs to be included.*

Response R1-1. We appreciate the valuable comments from Reviewer 1. As the reviewer suggested, we repeated the Delta neutralization assay and included Omicron BA.1 in the results of the neutralization assay (**Extended Data Fig. 2d; p6 lines 6-8; p7 lines 9-10; p19 lines 8-18**). We have to apologize that we could not expand to all SARS-CoV-2 variants due to the issue of limited institutional permission and capacity.

2) *The authors claim that SHM created more interactions between NCV2SG48 and the RBD. This could be tested by making germline reversion of the V-genes and running binding or neutralization assays. It would be useful to have an antibody sequence showing the sites of SHM and binding residues.*

Response R1-2. We thank the reviewer for suggesting a meaningful experiment. As the reviewer suggested, we reverted the V_H and V_L domains of the NCV2SG48 antibody to the germline sequence. Germline reverted NCV2SG48 antibody substantially reduced neutralization activity to Alpha, Beta, BA.1, BA.2, BA.2.12.1, and BA.4/5 but only minor reduction to Wuhan, D614G, and Delta indicating the contribution of SHM. These data are included in the revised manuscript (**Fig. 2b; Fig. 4e; p9 lines 4-12**).

3) *In Fig. 5a, determining the binding angle using the entire Fab is not appropriate. The Fv and Fc are connected by a flexible hinge and the conformation of the Fc in relation to the Fv in a crystal is likely more dependent on crystal contacts than biological relevant influences. This analysis should be limited to the angle of the Fv region. Also, please indicate how center of gravity was determined.*

Response R1-3. We thank the reviewer for pointing this out. The center of mass of the variable region (Heavy chain : residue 1 to 116, Light chain : residue 1-107) was recalculated and used to display the angular differences. The centers of mass were calculated using pymol's centerofmass script based on atomic positions only. According to this, **Fig. 5a** was revised.

4) *In the method for the convalescent or vaccinated human donors section, please clarify what the healthy volunteers were used for and the timing of the blood samples taken after vaccination. Are these the “uninfected” donors used in the study? I find it confusing if these are the uninfected controls, how do they show S specific IgM, IgG, and IgA levels? How were they confirmed to be uninfected?*

Response R1-4. We apologize for an unnecessary description of the method caused by deleting vaccinated serum data from the previous manuscript version. We corrected and clarified the healthy volunteers in the method section. All blood samples used in this study were collected before taking any SARS-CoV-2 vaccination. We confirmed uninfected/unvaccinated donors by their clinical history and ELISA titer. S-specific Ig titers were determined using serial serum dilution on S-trimer-coated wells next to Ig-capturing antibody standard wells on the same ELISA plate. Serum from negative donors generally show around the detection limit of S-specific IgG (< 100 ng/ml) and IgA (< 10 ng/ml) as seen in **Extended Data Fig. 1a** (p14 lines 1-9; p16 lines 20-22).

5) *All figures should include definitions of what error bars or box plots represent and the number of replicates. Fig. 1 and Extended Data Fig. 1 are missing this information.*

Response R1-5. According to the comment from the reviewer, we checked all figures and described an explanation of box plots, the number of replicates, and what data indicate (**Fig. 1a, e, f; Fig. 2a, b; Extended Data Fig. 1-3**).

Minor Revisions:

1) *The manuscript would benefit from some clarifying edits in addition to point 4 above. For example, the first two sentences of the introduction are redundant as they are restating nearly the same thing. The sentence starting with “Notably...” on page 8, line 9 was also confusing and required multiple reads to understand what was being described.*

Response R1-6. We thank the reviewer for pointing this out. We merged the two redundant sentences in the introduction (p2 lines 16-18). The sentence starting with “Notably...” was intensively revised describing it in detail (p8 lines 9-19).

2) *The Rfree test set for the NCV2SG48 Fab-wuhan-Hu-1 structure is only 3.7% of reflections instead of the typical 5%. Why is this? The other two structures both use 4.9% of reflections.*

Response R1-7. The reason for the 3.7% free flag was due to the fact that the free flag was not extended when high-resolution reflections were added during the structural refinement process. To properly enable cross-validation, new 5% free flags were re-set for the NCV2SG48 Fab-Wuhan-Hu-1 data, and the structural analysis was re-evaluated from phase determination using the same procedures. The figures and all values in the Tables were re-calculated by the newly determined structure. No significant changes in structure and calculated values were found. The new structure and that data were re-registered to PDB and the new validation reports were submitted.

3) *In Extended Data Fig. 2e-f, the column for the CDR3 sequences needs to be adjusted so that the letters are legible*

Response R1-8. We thank the reviewer for the comment. CDR3 sequences in **Extended Data Fig. 2e-f** are shown in larger letters in the revised manuscript.

4) *In the crystallization section of the methods, the units for the protein concentration should read mg/mL or mg mL⁻¹ with the -1 superscripted.*

Response R1-9. We thank the reviewer for the comment. We corrected the unit to mg/ml (**p21 lines 3-10**).

5) *Include PDB ID for the omicron structure in the Data and availability section.*

Response R1-10. We thank the reviewer for the comment. We included PDB IDs in the Data availability section for NCV2SG48-Wuhan, Delta, and Omicron BA.1; NCV2SG53-Wuhan and Delta (**p22 lines 13-19**).

Reviewer 2

1. *There were similar broadly neutralizing antibodies reported. The author should have a systematic comparison to these reported broadly neutralizing antibodies.*

Response R2-1. We appreciate the valuable comments from Reviewer 2. As suggested by the reviewer, we generated an additional figure summarizing the reported broadly neutralizing antibodies of EUA to compare with mAbs in this study (**Extended Data Fig. 4b**). According to this figure, we added discussion (**p12 lines 19-24**).

2. *Many broadly neutralizing antibodies showed neutralizing activity against BA.1, however, lost their neutralizing activity against BA.4 and BA.5. Could the author do additional assays on BA.4 or BA.5? or based on the complex structure, have a prediction on the neutralizing potential of NCV2SG48 against BA.4 or BA.5. Will the binding of NCV2SG48 be affected by the mutations of BA.4 or BA.5?*

Response R2-2. We thank the reviewer for suggesting a valuable experiment. We tested neutralization activity against BA.4/5 (BA.4 and BA.5 have the same S protein sequence) and additional Omicron variants. We found that NCV2SG48 maintains neutralizing activity to Omicron variants BA.2, BA.2.12.1, and BA.4/5 (**Fig. 2a,b; Extended Data Fig. 4b; p2 line 6; p7 lines 2-9**). This is further supported by the binding affinity assay (**Extended Data Fig. 3a**).

3. *An interesting point raised by the research is that the percentage of broadly neutralizing antibodies increases as the immune system has a longer interaction with the virus, which was supposed to be the results of accumulated somatic hypermutations. This could be very helpful in guiding the development of vaccines. However, the number of the samples is not big enough to make such a conclusion. Has this been observed by others? Or would data from similar research support this?*

Response R2-3. We appreciate the valuable comments from the reviewer. We included appropriate references about the importance of accumulated SHMs in the binding affinity to SARS-CoV-2 (**p12 lines 5-7**).

Reviewer 3

Major Notes

1. *Figure 1 and figure 2: The comparison should include other Omicron subvariants including BA.2, BA.2.12.1, and BA.4/BA.5.*

Response R3-1. We appreciate the valuable comments from Reviewer 3. We added data for this point. Because this point has been raised by Reviewer 2, please see our response R2-2.

2. *Line 20 page 6: Only prototype RBD was used in the binding affinity measurement of antibodies, which could not support in estimating the changes of affinity among different strains. Thus, the test on other Omicron subvariants would be helpful, see the above comment.*

Response R3-2. We thank the reviewer for the comment. To estimate the changes in affinity among different strains, we performed an additional binding affinity assay. The binding affinity of NCV2SG48 and NCV2SG53 mAbs was unchanged against Delta compared to Wuhan. However, we observed higher dissociation of NCV2SG48 mAb against Omicron variants, BA.1, BA.2, and BA.4/5 supporting reduced neutralization activity to Omicron subvariants. This data is included in the revised manuscript (Extended Data Fig. 3a).

3. *Figure 2b: The color scheme for the two curves of antibodies combination should be improve to distinguish them in far different color.*

Response R3-3. We thank the reviewer for the comment. Since we included additional Omicron variants BA.2, BA.1.12.1, and BA.4/5 in the analysis, Fig. 2 became busy showing all data by the line graphs. Therefore, we withdrew the previous Fig. 2b and summarized those with additional data in the table shown in the new Fig. 2b.

4. *Line 24 page 7: According to the previous report, antibodies cocktails contain two or more non-overlapping antibodies to enhance the neutralizing potency and decrease the potential for SARS-CoV-2 escape mutants. But the NCV2SG48 and NCV2SG53 recognize the partially overlapping epitopes, causing theoretically competitive binding to RBD between NCV2SG48 and NCV2SG53. So, the author should explain the possible reason for still highly neutralizing effectiveness at a low dose when using NCV2SG48 and NCV2SG53 cocktails? Are they synergetic in function?*

Response R3-4. We appreciate the valuable comments. As the reviewer pointed out, the structural analysis revealed that NCV2SG48 and NCV2SG53 recognize partially overlapped epitopes (Fig. 3c and Extended Data Fig. 5c), however, the mAb cocktail consisting of NCV2SG48 and NCV2SG53 acts in a complementary manner. Namely, this class 1/2 cocktail can neutralize SARS-CoV-2 variants with a lower IC₅₀ value without intercepting the other. This point has been included in the discussion (p12 line 26-p13 line 4).

5. *Figure 3: The two complex structures should be shown separately. You may add them to some appropriate place in Fig. 3.*

Response R3-5. According to the comment from the reviewer, we separated the structure model of NCV2SG48 and NCV2SG53 mAbs as the new Fig. 3b, c.

6. *The colors rendering of structural figures are confusing. The authors should keep consistent colors for models and footprints. And in the footprints, it should be revised to discriminate with similar colors as models.*

Response R3-6. We thank the reviewer for pointing this out. According to the reviewer's suggestion, we changed to consistent colors for the structural models, HC in blue and LC in green (Fig. 3d,e; Fig. 4c; Fig. 5c; Fig. 6b,c). Since the footprint in **Extended Data Fig. 4a** was too complicated and our purpose is just to compare binding patterns on RBD among mAbs, we changed only red in footprints to avoid confusion.

REVIEWERS' COMMENTS:

Reviewer #1 (Remarks to the Author):

The authors have addressed all of my concerns and I recommend this manuscript for publication.

Reviewer #2 (Remarks to the Author):

All the concerns have been addressed in the revised manuscript.

Reviewer #3 (Remarks to the Author):

This is a markedly improved version of the manuscript. Notably, new data were added on the Omicron subvariants neutralization and I believe that this makes the revised manuscript a much stronger candidate for this journal. However, to ensure the integrity of the data, the authors need to provide all the affinity kinetics curves tested rather than presenting only WT RBD.

Point-by-point responses (MS: COMMSBIO-22-2894A)

REVIEWERS' COMMENTS:

Reviewer #1 (Remarks to the Author): The authors have addressed all of my concerns and I recommend this manuscript for publication.

Response: We thank Reviewer #1 for valuable comments and suggestions.

Reviewer #2 (Remarks to the Author): All the concerns have been addressed in the revised manuscript.

Response: We thank Reviewer #2 for valuable comments and suggestions.

Reviewer #3 (Remarks to the Author):

This is a markedly improved version of the manuscript. Notably, new data were added on the Omicron subvariants neutralization and I believe that this makes the revised manuscript a much stronger candidate for this journal. However, to ensure the integrity of the data, the authors need to provide all the affinity kinetics curves tested rather than presenting only WT RBD.

Response: We thank Reviewer #3 for valuable comments and suggestions. According to the reviewer's request, we included all the affinity kinetics curves in addition to the summary table in Supplementary Figure 3.